# pH-dependent 11° F₁FO ATP synthase sub-steps reveal insight into the FO torque generating mechanism

**Seiga Yanagisawa, Wayne D Frasch***

School of Life Sciences, Arizona State University, Tempe, United States

**Abstract** Most cellular ATP is made by rotary $F_1F_O$ ATP synthases using proton translocation-generated clockwise torque on the $F_O$ c-ring rotor, while $F_1$-ATP hydrolysis can force counterclockwise rotation and proton pumping. The $F_O$ torque-generating mechanism remains elusive even though the $F_O$ interface of stator subunit-a, which contains the transmembrane proton half-channels, and the c-ring is known from recent $F_1F_O$ structures. Here, single-molecule $F_1F_O$ rotation studies determined that the pKa values of the half-channels differ, show that mutations of residues in these channels change the pKa values of both half-channels, and reveal the ability of $F_O$ to undergo single c-subunit rotational stepping. These experiments provide evidence to support the hypothesis that proton translocation through $F_O$ operates via a Grotthuss mechanism involving a column of single water molecules in each half-channel linked by proton translocation-dependent c-ring rotation. We also observed pH-dependent 11° ATP synthase-direction sub-steps of the *Escherichia coli* $c_{10}$-ring of $F_1F_O$ against the torque of $F_1$-ATPase-dependent rotation that result from $H^+$ transfer events from $F_O$ subunit-a groups with a low pKa to one c-subunit in the c-ring, and from an adjacent c-subunit to stator groups with a high pKa. These results support a mechanism in which alternating proton translocation-dependent 11° and 25° synthase-direction rotational sub-steps of the $c_{10}$-ring occur to sustain $F_1F_O$ ATP synthesis.

**\*For correspondence:**
frasch@asu.edu

**Competing interest:** The authors declare that no competing interests exist.

## Editor's evaluation

This manuscript describes single molecule spectroscopic work to probe the mechanisms by which protonation/deprotonation steps produce torque between the a-subunit and the C10 ring, which is subsequently conveyed to F1 to couple to ATP synthesis. This is an important bioenergetics question and the approach yields some tantalizing clues as to which protonation steps are involved. In principle, this knowledge could lead to direct experimental or computational tests to resolve the overall mechanism. The previous issues in the manuscript were well addressed by the authors in the new revision. The reviewers found a few small additional issues, which the authors can address in the final revisions.

## Introduction

The $F_1F_O$ ATP synthase (**Figure 1**) that is found in all animals, plants, and eubacteria is comprised of two molecular motors that are attached by their rotors and by their stators (**Kühlbrandt, 2019**; **Spetzler et al., 2012**). The $F_O$ motor, which is embedded in bioenergetic membranes, uses a non-equilibrium transmembrane chemiosmotic proton gradient also known as a proton-motive force (pmf) to power clockwise (CW) rotation of its ring of c-subunits relative to the subunit-a and -b stator proteins as viewed from the *Escherichia coli* periplasm. These subunits contribute to the peripheral stalk bound to one side of the $F_1$ $(\alpha\beta)_3$-subunit ring where each αβ-heterodimer comprises a catalytic site that

**Figure 1.** Cryo-EM structures of $F_1F_O$ ATP synthase inhibited by ADP in three rotary states, and measurement of changes in rotational position between catalytic dwells. (**A**) Rotational state-1, pdb-ID 6OQU (*Sobti et al., 2020*). (**B**) State-2, pdb-ID 6OQV, with rotor 120° counterclockwise (CCW) from (**A**) where subunit-α is not shown to reveal subunit-γ. (**C**) State-3, pdb-ID 6WNR, with rotor 240° CCW from (**A**) showing microscope slide assembly of $F_1F_O$ embedded in a lipid bilayer nanodisc (LBN) for rotation measurements. $His_6$-tags (HT) on β-subunit C-termini enabled attachment to slide, while the gold nanorod (AuNR) coated with streptavidin (SA) bound to the biotinylated subunit c-ring. (**D**) Rotational position of single $F_1F_O$ molecules versus time was monitored by intensity changes of polarized red light scattered from the AuNR in the presence of 1 mM $Mg^{2+}$ATP, which enabled $F_1$-ATPase-dependent 120° CCW power strokes between catalytic dwells (green bars). Prior to data collection at 200 kHz, a polarizer in the scattered light path was rotated to minimize intensity during one of the three catalytic dwells. Light intensity increased to a maximum upon rotation by 90° during the subsequent CCW 120° power stroke. For each molecule the angular dependence of these power strokes versus time was analyzed.

synthesizes ATP from ADP and Pi. Subunit-$\gamma$, which docks to the c-ring along with subunit-$\varepsilon$, extends into the core of the $(\alpha\beta)_3$-ring (*Figure 1B*). The rotary position of subunit-$\gamma$ within the $(\alpha\beta)_3$-ring causes the conformations of the three catalytic sites to differ such that one site contains ADP and Pi, a second site contains ATP, and the third site is empty. Pmf-powered CW rotation of subunit-$\gamma$ forces conformational changes to all catalytic sites in the $(\alpha\beta)_3$-ring, which releases ATP from one catalytic site with each 120° rotational step (*Kühlbrandt, 2019*; *Spetzler et al., 2012*). In this manner, $F_1F_O$ converts the energy from the pmf ($\Delta\mu_H+$) into energy in the form of a non-equilibrium chemical gradient ($\Delta\mu_{ATP}$) where the ATP/ADP•Pi concentration ratio is far in excess of that found at equilibrium.

When $\Delta G_{ATP}$ is significantly higher than n*pmf, the $F_1$-ATPase motor can overpower the $F_O$ motor and catalyze net ATP hydrolysis (*Steigmiller et al., 2008*; *Fischer et al., 2000*). This results in ATPase-dependent power strokes that rotate continuously CCW for 120° at saturating ATP concentrations (*Yasuda et al., 2001*; *Spetzler et al., 2006*), and pump protons across the membrane to the periplasm in *E. coli* (*Spetzler et al., 2012*). Power strokes are separated by catalytic dwells that last a few ms, during which ATP is hydrolyzed (*Yasuda et al., 2001*; *Spetzler et al., 2006*; *Spetzler et al., 2009*). At rate-limiting ATP concentrations, an ATP-binding dwell can interrupt the *E. coli* $F_1$ power stroke ~34° after the catalytic dwell (*Yasuda et al., 2001*; *Martin et al., 2014*). At this same rotary position, high ADP concentrations can compete with ATP to bind to the empty catalytic site resulting in ADP inhibition (*Martin et al., 2014*). All ATP synthases can catalyze ATP hydrolysis to some extent, although many have evolved regulatory mechanisms to minimize this energy wasteful process (*Kühlbrandt, 2019*). Under some circumstances, *E. coli* employs $F_1F_O$ as an ATPase-driven $H^+$ pump to maintain a pmf as an energy source for other metabolic processes (*Spetzler et al., 2012*).

The means by which $H^+$ translocation from the periplasm generates CW rotational torque on the c-ring is poorly understood. Molecular motors are believed to operate by either power stroke or by a Brownian ratchet mechanism, which has been postulated for $F_O$ (*Hwang and Karplus, 2019*; *Oster et al., 2000*). Although evidence clearly supports a power stroke mechanism for $F_1$-ATPase-dependent rotation (*Martin et al., 2014*; *Martin et al., 2018*; *Pu and Karplus, 2008*), there is little direct evidence in support of either mechanism for $F_O$-driven rotation in the ATP synthase direction. Protons enter and exit $F_O$ via half-channels in stator subunit-a. Two c-subunits in the ring contact subunit-a at a time where the leading cD61 during CW rotation (synthase direction) accepts a proton from the input channel, while the lagging cD61 donates its proton to the output channel.

The half-channels are separated by the highly conserved subunit-a arginine (aR210 in *E. coli*), which has long been thought to be responsible for displacing the $H^+$ from cD61 into the output channel during ATP synthesis (*Lightowlers et al., 1987*; *Cain and Simoni, 1989*; *Cain and Simoni, 1988*; *Vik et al., 1988*). As the result of mutations of *E. coli* subunit-a residues aN214, aE219, aH245, aQ252, and aE196 to other groups including leucine that decreased ATP synthase activity, ATPase-dependent $H^+$ pumping, and altered the ATP hydrolysis activity, these residues were proposed to translocate protons directly along the half-channels (*Lightowlers et al., 1988*; *Howitt et al., 1990*; *Eya et al., 1991*; *Hartzog and Cain, 1994*; *Hatch et al., 1995*; *Hahn et al., 2018*). Cryo-EM $F_1F_O$ structures that reveal details of subunit-a confirmed that these residues are positioned along possible half-channels that are separated by aR210 (*Hahn et al., 2018*; *Zhou et al., 2015*; *Pinke et al., 2020*; *Sobti et al., 2020*; *Martin et al., 2015*).

Alternatively, $H^+$ translocation through $F_O$ has also been postulated to occur via a Grotthuss mechanism (*Cukierman, 2006*) where a column of single water molecules that are hydrogen-bonded to specific protein groups behave in a coherent manner to transfer protonic charge over long distances via rapid exchange of $H^+$ between $H_3O$ and $H_2O$ (*Cukierman, 2006*; *Wraight, 2006*). A recent $F_1F_O$ structure from bovine mitochondria was of sufficient resolution to observe density near the input channel residues consistent with Grotthuss-type water molecules in this half-channel (*Spikes et al., 2020*). A consequence of this coherent behavior of a Grotthuss water column is that the rate of $H^+$ transfer can be much faster than the rate via free diffusion (*Cukierman, 2006*). The possibility that $F_O$ operates via a Grotthuss mechanism was first suggested from the observation of an astounding $H^+$ translocation rate of 6240 $H^+$ $s^{-1}$ from a driving force of 100 mV across *Rhodobacter capsulatus* vesicles containing $F_O$ that lacked $F_1$ (*Feniouk et al., 2004*). The *R. capsulatus* $F_O$ rates of $H^+$ transfer exceed the rate of delivery of protons by free diffusion from the bulk aqueous solution at a concentration of $10^{-8}$ M (pH 8) such that the ability to supply protons to a Grotthuss water column should be rate-limiting (*Wraight, 2006*). To achieve this rate of $H^+$ translocation, the existence of a $H^+$ antenna at the

entrance to the $F_O$ input channel has been postulated (*Wraight, 2006*), which in *R. capsulatus*, was calculated to consist of a hemispherical Coulomb cage with a $H^+$ capture radius of ~40 Å surrounding the entrance to the input channel.

Single-molecule studies of $F_1F_O$ molecules embedded in lipid bilayer nanodiscs (*Figure 1C*) revealed that the 120° CCW ATPase power strokes can be interrupted by transient dwells (TDs) at ~36° intervals with a duration of ~150 μs (*Martin et al., 2015*; *Ishmukhametov et al., 2010*; *Yanagisawa and Frasch, 2017*). In more than 70% of TDs, the $F_O$ motor not only halted $F_1$-ATPase CCW rotation, but the c-ring rotated CW in synthase-direction steps (*Martin et al., 2015*; *Yanagisawa and Frasch, 2017*). Complete assembly of $F_1F_O$ nanodisc complexes from the membrane scaffold protein (MSP), lipids, and detergent-solubilized $F_1F_O$ was verified by 2D electrophoresis where the first nondenaturing gel dimension contained a single band, which after the second denaturing dimension, contained bands corresponding to MSP and all $F_1F_O$ subunits (*Ishmukhametov et al., 2010*). The ATPase activity of nanodisc preparations is DCCD-sensitive, remains unchanged after 8 hr at room temperature, and is 1.5-fold higher than the initial activity of detergent-solubilized $F_1F_O$, which loses all activity in <4 hr. The observation that a subunit-a insertion mutation, which disrupts the interface between subunit-a and the c-ring, retained ATPase activity but lost the ability to form TDs indicates that these dwells result from an interaction between subunit-a and successive c-subunits in the c-ring (*Ishmukhametov et al., 2010*).

The occurrence of TDs was found to increase at pH 8 when viscous drag on the nanorod is sufficient to slow the angular velocity of the $F_1$ ATPase-driven power stroke (*Martin et al., 2015*; *Ishmukhametov et al., 2010*). These results showed that there is a kinetic component that affects the probability that the interaction between subunit-a and the c-ring will occur relative to the $F_1$-ATPase power stroke. A kinetic dependence for $F_1F_O$-catalyzed ATP synthesis versus hydrolysis has been theorized based on energetic calculations (*Gao et al., 2005*). Occurrence of TDs, including those with synthase-direction steps, is also known to increase inversely with pH between pH 5 and pH 7 (*Yanagisawa and Frasch, 2017*). This suggests that synthase-direction steps depend on $H^+$ transfer from the protonated groups with a low pKa from the subunit-a input channel to the c-ring, and from the c-ring to unprotonated groups with a high pKa in subunit-a output channel.

Maximal ATP synthase rates in membrane vesicles where *E. coli* $F_1F_O$ is oriented with $F_1$ on the outer surface are typically achieved with inner and outer pH values of 5.0 and 8.5, respectively (*Fischer et al., 2000*; *Fischer et al., 1994*). The pmf can be derived from non-equilibrium differences in pH ($\Delta$pH) or membrane potential ($\Delta \phi$). Although the energy from $\Delta$pH and $\Delta \phi$ to drive ATP synthesis are interconvertible, the latter is the dominant energy source for *E. coli* in vivo.

Each of the three successive 120° $F_1$-ATPase power strokes required for a full revolution of the $F_1F_O$ rotor is unique because the rotary positions of subunit-γ differ relative to the peripheral stalk that includes subunit-a (*Figure 1A–C*). These power strokes also differ because the 36° stepping of the $c_{10}$-ring and the 120° power strokes can only be aligned during one of the three catalytic dwells. As a result, the three power strokes require the translocation of 4 $H^+$, 3 $H^+$, and 3 $H^+$ that result in net c-ring rotations of 144°, 108°, and 108°, respectively. Consequently, the c-ring and subunit-γ become misaligned by +14° and −14° during two of the catalytic dwells. The elasticities of the peripheral stalk, subunit-δ, and to some extent, subunit-γ accommodate these rotary differences (*Sobti et al., 2020*; *Murphy et al., 2019*).

The positive and negative torsion on the $c_{10}$-ring from the elastic energy needed to accommodate the +14° and −14° misalignments during rotation affects the ability to form TDs and their associated synthase-direction steps (*Yanagisawa and Frasch, 2017*), which along with their pH dependence and occurrence every 36°, indicate that they correspond to single c-subunit stepping relative to subunit-a. To determine the percentage of power strokes in which TDs were observed, data sets were collected from one of the three 120° power strokes from each single-molecule of nanodisc-embedded *E. coli* $F_1F_O$ examined. For each data set that comprised ~300 power strokes, the percent of power strokes that contained TDs was determined. For each molecule examined there was an equal chance that the c-ring and catalytic dwell was aligned, or subject to the positive or negative torsion from misalignment. The distribution of data sets collected from many molecules versus the percent of power strokes containing TDs fit to the sum of three Gaussians that corresponded to low, medium, and high probabilities of TD formation. The high and low TD percentages were consistent with the torsion from misalignment that provides additional energy to promote TD formation, or to inhibit it when torsion is

in the opposite direction (*Yanagisawa and Frasch, 2017*). Similar effects of the $c_{10}$-ring and catalytic dwell misalignments have also been observed in other single-molecule $F_1F_O$ measurements (*Sielaff et al., 2019*).

We have now examined mutations of residues in the putative subunit-a half-channels that have been implicated to participate in $H^+$ transfer events in both the input and output channels with the goal of distinguishing whether $F_O$ rotation occurs via a Brownian ratchet versus a power stroke mechanism, and whether the half-channel residues transfer protons individually or act together to support water column that transfers protons via a Grotthuss mechanism.

The ability to form ATP synthase-direction steps as a function of pH during ATP hydrolysis-driven CCW power strokes was characterized in single-molecule studies of $F_1F_O$ molecules embedded in lipid bilayer nanodiscs. Formation of synthase-direction steps during a TD was maximal at the pH value when unprotonated and protonated forms of the input and output channels were optimal such that synthase-direction steps required $H^+$ transfer both from the input channel to the leading c-subunit and from the lagging c-subunit to the output channel. Mutation of a residue from either the input or output channel altered both the low and high pKa values of TD formation indicating that input and output channels communicate. This is consistent with a Grotthuss mechanism where a water column in each of the two channels is connected by rotation-dependent $H^+$ transfer to and from the leading and lagging c-subunits in the c-ring, which is also supported by features from a variety of $F_1F_O$ structures.

The extent of rotation during ATP synthase-direction steps was unexpectedly found to rotate 11° in the WT and all mutants. Cryo-EM structures of sub-states with subunit-a:c-ring differences of 11° that position the lagging c-subunit cD61 adjacent to output residues aS199 and aE196 (*Sobti et al., 2020*) are consistent with the synthase-direction steps observed here. When combined with structural information, the results do not support the hypothesis that the role of aR210 is to displace the $H^+$ from cD61, but are consistent with a Grotthuss $H^+$ translocation mechanism involving both half-channels for sustained ATP synthase-direction c-ring rotation that results from successive alternating 11° and 25° synthase-direction sub-steps for each c-subunit in the $c_{10}$-ring. Direct evidence of a mixed model was observed in which some synthase-direction steps show characteristics of a power stroke while others exhibit oscillations consistent with a Brownian ratchet mechanism.

## Results

Contributions of subunit-a residues putatively involved in the ATP synthase $H^+$ half-channels were assessed by the effects on TD formation caused by mutations that converted charged or polar groups in subunit-a to hydrophobic leucine. Changes in rotational position were measured by a 35 × 75 nm gold nanorod (AuNR) bound to the biotinylated c-ring of individual *E. coli* $F_OF_1$ molecules embedded in lipid bilayer nanodiscs (*Ishmukhametov et al., 2010*), hereafter $F_1F_O$ (*Figure 1C*). Changes in rotational position during $F_1$-ATPase power strokes in the presence of saturating 1 mM MgATP were monitored by the intensity of polarized red light scattered from the AuNR (*Spetzler et al., 2006*; *Hornung et al., 2011*). Prior to data collection, the polarizer was adjusted so that the scattered red light intensity was at a minimum during one of the three $F_1$ catalytic dwells (*Figures 1D and 2A*). The subsequent power stroke caused an increase in light intensity to a maximum when the AuNR had rotated 90° (*Ragunathan et al., 2017*). Rotational data sets of each $F_1F_O$ molecule examined were collected for 5 s, which included ~300 of these power strokes (*Yanagisawa and Frasch, 2017*). Ten data sets were collected for each molecule. The number of $F_1F_O$ molecules examined at each pH for WT and mutants is indicated in *Figure 2—figure supplement 3*. Using WT at pH 5.0 as an example where data from 103 $F_1F_O$ molecules were collected, this was equivalent to 1030 data sets, and ~309,000 power strokes examined. For each molecule examined, rotational position versus time was calculated from scattered light intensity versus time using an arcsine$^{1/2}$ function from which the number of TDs observed during the first 90° of rotation were determined (*Ragunathan et al., 2017*).

Example power strokes from WT and mutant $F_1F_O$ molecules at pH 5.0 where TDs were present (black dots) and absent (blue dots) are shown in *Figure 2A* and *Figure 2—figure supplement 1*, respectively. When present, TDs either stopped $F_1$-ATPase CCW rotation momentarily (green dots) or exhibited CW rotation in the ATP synthase direction, hereafter synthase-direction steps (red dots). None of the mutations examined eliminated the ability of $F_1F_O$ to form TDs. Power strokes typically contained two to three TDs, when present. These were separated by an average of ~36°, consistent with an interaction between subunit-a and successive c-subunits in the $c_{10}$-ring of *E. coli* $F_1F_O$.

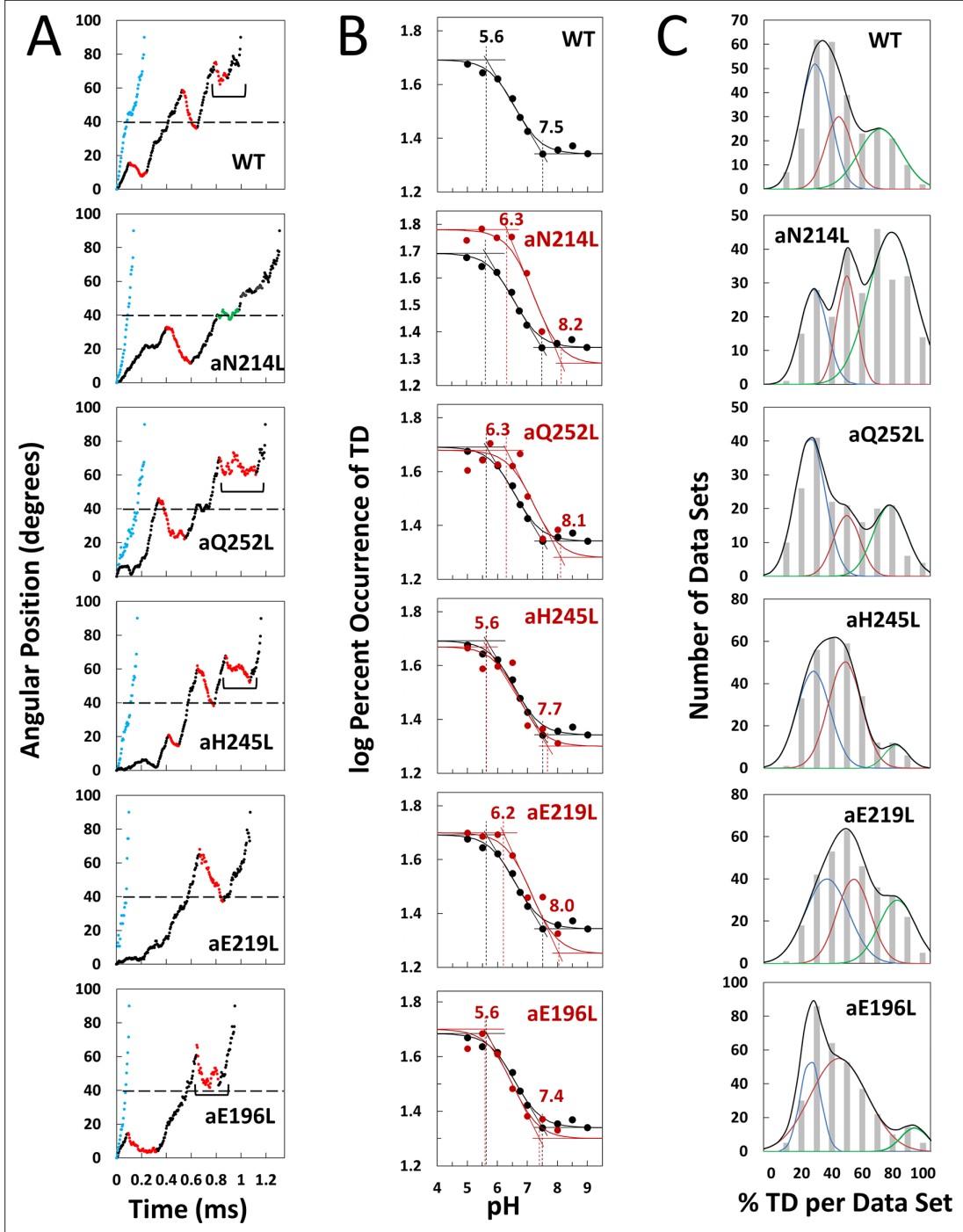

**Figure 2.** Effects of subunit-a mutations on transient dwells (TDs). (**A**) Examples of power strokes without TDs (blue), and of power strokes with TDs that lacked (green), or contained clockwise (CW) synthase-direction c-ring rotation relative to subunit-a (red) plotted as degrees of rotation after the catalytic dwell versus time where 40° (dotted line) is the optimal position for binding of ATP or inhibitory ADP (*Yasuda et al., 2001*; *Martin et al., 2014*). Brackets indicate Brownian-type oscillations during a TD. (**B**) Average percent TDs per data set versus pH from which pKa values were derived via intercepts of the slope and plateaus (solid line) of each curve based on the fit of the data to *Equation 1* for WT (black line) and subunit-a mutants (red line). (**C**) Distributions at pH 6.0 of the percent of TDs per data set of power strokes (gray bars) where multiple data sets that each contained ~300 power strokes were collected from each of the total number of the $F_1F_O$ indicated, and data were binned in 10% increments. The data were fit to the sum of three Gaussians (black line) representing low (blue), medium (orange), and high (green) efficiencies of TD formation.

The online version of this article includes the following figure supplement(s) for figure 2:

**Figure supplement 1.** Examples each of the first 90° of ATP hydrolysis-driven power strokes observed using $F_OF_1$ nanodiscs.

*Figure 2 continued on next page*

*Figure 2 continued*

**Figure supplement 2.** Examples of how changes in the variables in *Equation 1* affect the log-log plots that describe the F$_1$-ATPase inhibition kinetics of *Figure 2C*.

**Figure supplement 3.** Distribution of power stroke data sets (each set containing ~300 power strokes) at each pH examined versus the percentage of the occurrence of transient dwells (TDs) per data set binned to each 10% (gray bars).

A power stroke mechanism has been defined as the generation of a large free energy gradient over a distance comparable to the step size of the molecular motion so that transition to the forward position occurs nearly irreversibly (*Hwang and Karplus, 2019*). By contrast, in a Brownian ratchet mechanism the motor is thought to visit previous and forward positions through thermal motion, where stabilization in the forward position results by conformational changes triggered by the fuel processing event. While some synthase-direction steps shown of *Figure 2A* and *Figure 2—figure supplement 1* rotated CW in a concerted, and apparently irreversible manner characteristic of a power stroke, others indicated by brackets were observed to oscillate back and forth during the TD. These oscillations most commonly occurred late in the F$_1$ power stroke (~70–80°) and were more pronounced in all mutations examined except aN214L (*Figure 2—figure supplement 1*). Such oscillations are direct evidence of a Brownian ratchet mechanism and are likely the result of a close balance between the energy that powers the F$_1$-ATPases power stroke with the energy that powers synthase-direction rotation, which suggests that these mutations cause a decrease in the energy to power synthase-direction rotation.

## Subunit-a mutations alter pKas of TD formation

We postulated that mutation of subunit-a residues involved in H$^+$ translocation related to c-ring rotation would alter the pKa of the half-channel in which the is located if each residue contributes independently to the H$^+$ translocation process. Consequently, we precisely determined the pKa values of groups that contribute to TD formation (*Figure 2B*) using equations applied to the pH dependence of enzyme inhibition kinetics (*Cook and Cleland, 2007*).

TDs occur when subunit-a binds to the c-ring to stop F$_1$ ATPase-driven rotation for a period of time. Thus, a TD represents an extent that F$_O$ inhibited the F$_1$ATPase motor, which occur as often as 3.6 times per F$_1$ power stroke. Kinetically, the ATPase power stroke duration without TDs is ~200 μs, while the average duration of each TD is ~150 μs (*Martin et al., 2015*; *Ishmukhametov et al., 2010*; *Yanagisawa and Frasch, 2017*). In data sets where TDs occur in 100% of the power strokes, for example, aN214L at pH 6.0, all of the ~300 power strokes in that data set will contain TDs (i.e. they look like the power strokes in *Figure 2A* where TDs are denoted by green or red dots) such that the efficiency of TD formation is 100%. Such a data set represents a 64% inhibition of the F$_1$ATPase power stroke kinetics. When all ~300 power strokes in a data set look like those in *Figure 2A* represented by blue dots, the efficiency of TD formation in that data set is 0%.

Precise pKa determination depended upon the fits of the efficiency at each pH examined to the pH dependence over the range of pH values in *Figure 2B*. The number of molecules examined for WT and each mutant is indicated in *Figure 2—figure supplement 3*. For example, a total of 553 F$_1$F$_O$ molecules (~92 million power strokes) were examined to establish the pH dependence of WT. A maximum average of 47.5% of WT power strokes from all three efficiency groups occurred at pH 5.0, which decreased with increasing pH until it plateaued at a minimum of ~22% at pH values > 7.5 (*Figure 2B*). The pH dependences for WT and mutants were fit to *Equation 1* where T is the total average TD occurrence, T$_{min}$ is the minimum TD occurrence, and K$_1$ and K$_2$ are the inhibition constants that define the increase and maximum TD occurrence versus pH as the result of either a residue that is protonated with pKa$_1$, or unprotonated with pKa$_2$, respectively. It is noteworthy that K$_1$ is similar to a dissociation constant because a smaller K$_1$ increases the ability of subunit-a to

**Table 1.** pKa values and inhibition constants for WT and subunit-a mutants.

Values were derived from the fits to *Equation 1* of the average percent of TDs per data set versus pH in *Figure 2C*.

| | K$_1$ | K$_2$ | T$_{min}$ (%) | pK$_{a1}$ | pK$_{a2}$ |
|---|---|---|---|---|---|
| WT | 6.4 | 6.75 | 22.0 | 5.6 | 7.5 |
| aN214L | 7.0 | 7.50 | 19.1 | 6.3 | 8.2 |
| aQ252L | 7.0 | 7.40 | 19.5 | 6.3 | 8.1 |
| aE219L | 6.9 | 7.35 | 17.8 | 6.2 | 8.0 |
| aH245L | 6.5 | 6.87 | 20.0 | 5.6 | 7.7 |
| aE196L | 6.3 | 6.70 | 20.0 | 5.6 | 7.4 |

bind to, and stop, c-ring rotation with decreasing pH (*Figure 2—figure supplement 2*). Conversely, a smaller $K_2$ value decreases TD formation with decreasing pH because it is the unprotonated form of that residue that binds and inhibits.

$$T = \log T_{\min} - \log \left( 1 + \tfrac{K_1}{[H^+]} \right) + \log \left( 1 + \tfrac{K_2}{[H^+]} \right) \qquad (1)$$

The fit of the data to *Equation 1* defines the slope of the curve as well as the high and low plateau values. Because these are log-log plots, the pKa values (*Figure 2B*, dotted lines) are determined by the intercept of the slope with the high and low plateau values (solid lines). None of the mutations changed $T_{\min}$ significantly. Using parameters derived by the fits of the data to *Equation 1* for WT and mutants (*Table 1*), the WT group(s) that must be protonated to induce a TD had pKa$_1$ and $K_1$ of 5.6, and 6.4, respectively, while the group(s) that must be unprotonated to induce a TD had pKa$_2$ and $K_2$ values of 7.5 and 6.75, respectively.

The aN214L mutation, which had the greatest effect on the pH dependence of TD formation, increased the maximum percent of TDs formed at low pH to 61% (1.3-fold) and shifted the pH dependence in the alkaline direction from WT. These changes were due to increases in $K_1$ and $K_2$ to 6.4 and 6.75, respectively, that increased pKa$_1$ and pKa$_2$ by 0.9 and 0.7 pH units. The differential increases in $K_1$ and $K_2$ by 0.6 and 0.75 units led to the aN214L-dependent increase in maximum TD formation at low pH because an equal shift of these values in the same direction causes the curve to shift to higher pH values without affecting the maximum occurrence of TDs formed (*Figure 2—figure supplement 2*). Similar but smaller effects were observed with aQ252L and aE219L (*Figure 2B*) where $K_1$ increased by 0.6 and 0.5 units, respectively, resulting in a pKa$_1$ increase of almost 1 pH unit from that of WT. However, aQ252L and aE219L decreased $K_2$ by 0.35 and 0.40 units from WT such that the increase in pKa$_2$ was proportionally smaller than that observed for aN214L. Consequently, while both mutants shifted the pH dependence in the alkaline direction from that of WT, only aQ252L showed an increase in the maximum TD occurrence (52%).

Mutations aH245L and aE196L caused the smallest changes in the pH dependence of TD formation. The former increased $K_1$ and $K_2$ by 0.1 and 0.12 units, which had no effect on pKa$_1$, and increased pKa$_2$ by 0.2 units. The latter was the only mutation to decrease the values of both $K_1$ and $K_2$, which decreased pKa$_2$ by 0.1 pH units from that of WT. It is noteworthy that aE196 is a component of the H$^+$ output channel.

In summary, mutation of residues examined in either the input or the output channel changed both the high and low pKa values of TD formation. These results support the conclusion that they participate in H$^+$ translocation as first proposed from ensemble studies (*Lightowlers et al., 1987*; *Cain and Simoni, 1989*; *Cain and Simoni, 1988*; *Vik et al., 1988*; *Lightowlers et al., 1988*; *Howitt et al., 1990*; *Eya et al., 1991*; *Hartzog and Cain, 1994*; *Hatch et al., 1995*), but also show that the two channels communicate via the c-ring, which would occur if the residues in both channels support a Grotthuss-type water column connected by c-ring rotation.

## Subunit-a mutations affect TD formation efficiency

The percent of TDs observed per data set was fit to three Gaussian distributions with low (blue), medium (orange), and high (green) efficiencies as shown at pH 6.0 (*Figure 2C*), and at all pH values examined (*Figure 2—figure supplement 3*). These efficiency differences were proposed to result from elastic energy resulting from the 14° rotational mismatch between two of the three catalytic dwells and the c$_{10}$-ring that supplements or subtracts from the binding energy required for subunit-a to stop F$_1$-ATPase-driven rotation momentarily, resulting in a TD. If TDs result from H$^+$ translocation-dependent interactions between subunit-a and the c-ring, mutations that impact H$^+$ translocation should alter the TD formation efficiency.

Subunit-a mutations affected the percent of TDs formed per data set during power strokes in each of these efficiencies, which correlate to the three rotary positions of the central stalk relative to the peripheral stalk (*Yanagisawa and Frasch, 2017*; *Sielaff et al., 2019*). The proportional differences of efficiencies of TD formation are shown relative to the average low efficiency for WT (*Figure 3*). Medium and high efficiency distributions of TDs in WT increased 1.5-fold and 2.2-fold, respectively, relative to low efficiency. The aN214L mutation increased the percent of TDs per data set for high, medium, and low efficiencies by 3-fold, 2-fold, and 1.2-fold, respectively, from the WT low efficiency.

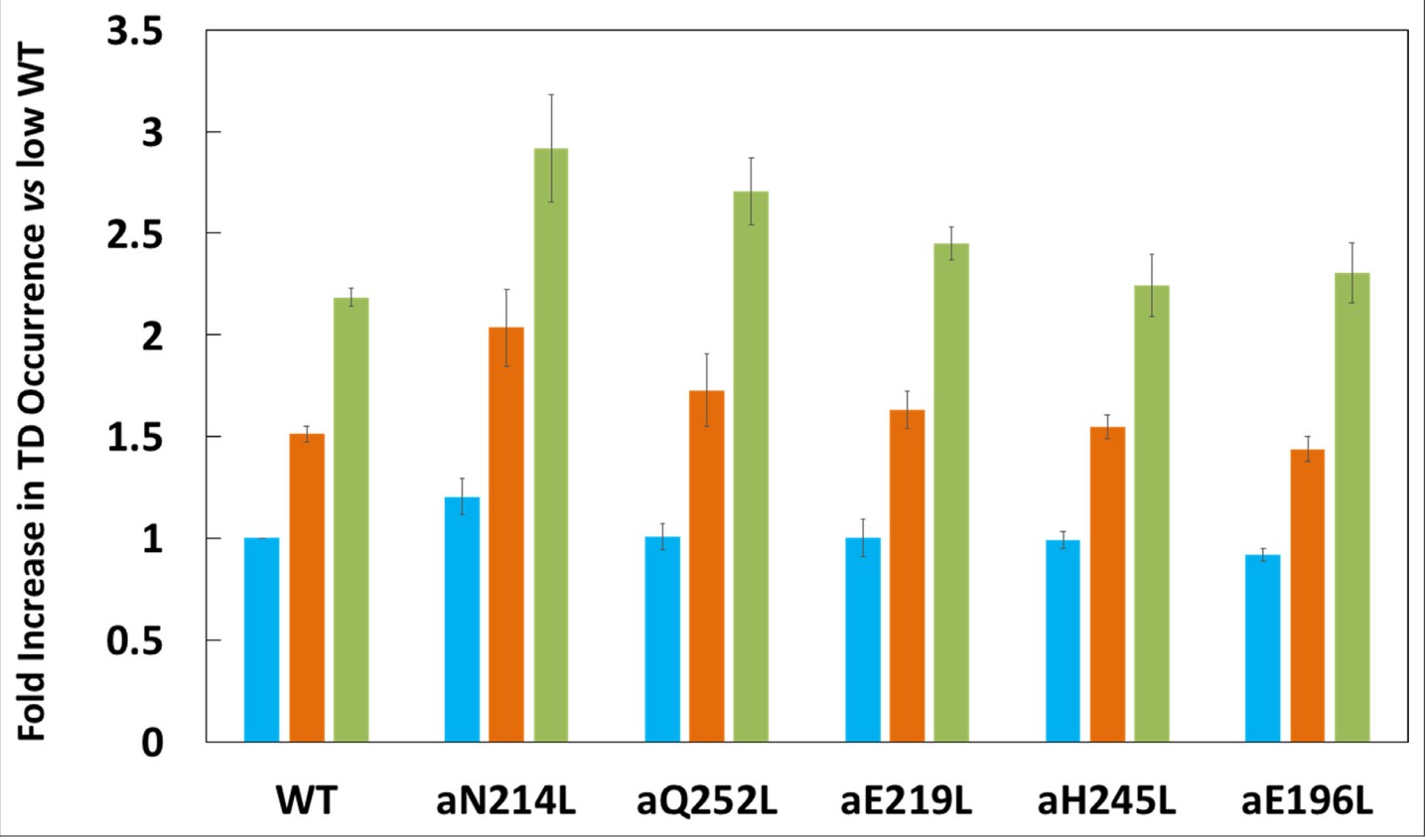

**Figure 3.** Proportion of low (blue bar), medium (orange bar), and high (green bar) transient dwell (TD) formation efficiencies relative to WT low efficiency TD formation.
 Each was the average of all pH values examined (*Figure 2—figure supplement 3*). All of the low, medium, and high efficiencies of TD formation are shown in that supplementary figure from which the averages were taken to calculate Figure 3. Vertical bars represent standard error.

Mutations aQ252L and aE219L also increased TDs per data set for the high (2.7-fold and 2.5-fold) and medium (1.7-fold and 1.6-fold), but not the low efficiency distributions. Mutations aH245L and aE196L either did not increase the efficiency or slightly decreased the efficiency of the distributions of TD formation per data set.

### Synthase-direction steps rotate CW an average of ~11°

The proportion of TDs with and without a synthase-direction step for WT and mutants are shown in *Figure 4A* at the pH values when the proportion of synthase-direction steps was minimum (black bars) and maximum (red bars), and at all pH values examined in *Figure 4—figure supplement 1*. The minimum proportion of synthase-direction steps was observed at pH 5.5 for WT and all mutants except aN214L that occurred at pH 6.0. Even at these low pH values, synthase-direction steps accounted for 62–68% of all TDs. In WT, a maximum of ~80% of TDs contained synthase-direction steps at pH 7.0, which was an increase of 13% from the minimum. These plots also show the distributions of the extent of CW rotation during a synthase-direction step, for which the 11° and 9° average and median values of CW rotation, respectively, were not changed significantly by the mutations (*Figure 4B*).

After subtracting the occurrence of the extent of synthase-direction step CW rotation at the pH when it was at a minimum (black bars) from that observed at other pH values (*Figure 4—figure supplement 1*) including that at its maximum (red bars), a Gaussian distribution of the increase in the extent of synthase-direction step CW rotation was observed (*Figure 4C*). During a synthase-direction step, the mean and standard deviations in the extent of CW rotation (*Figure 4—figure supplement 2*) was 12° ± 3° for WT, with little variation resulting from the mutations including: 11° ± 3° (aN214L), 11° ± 4° (aQ252L), 11° ± 3° (aH245L), 10° ± 3° (aE219L), and 11° ± 3° (aE196L) . In all cases, the distributions were truncated with minimum CW rotational steps of 6°. At their maxima, the extents

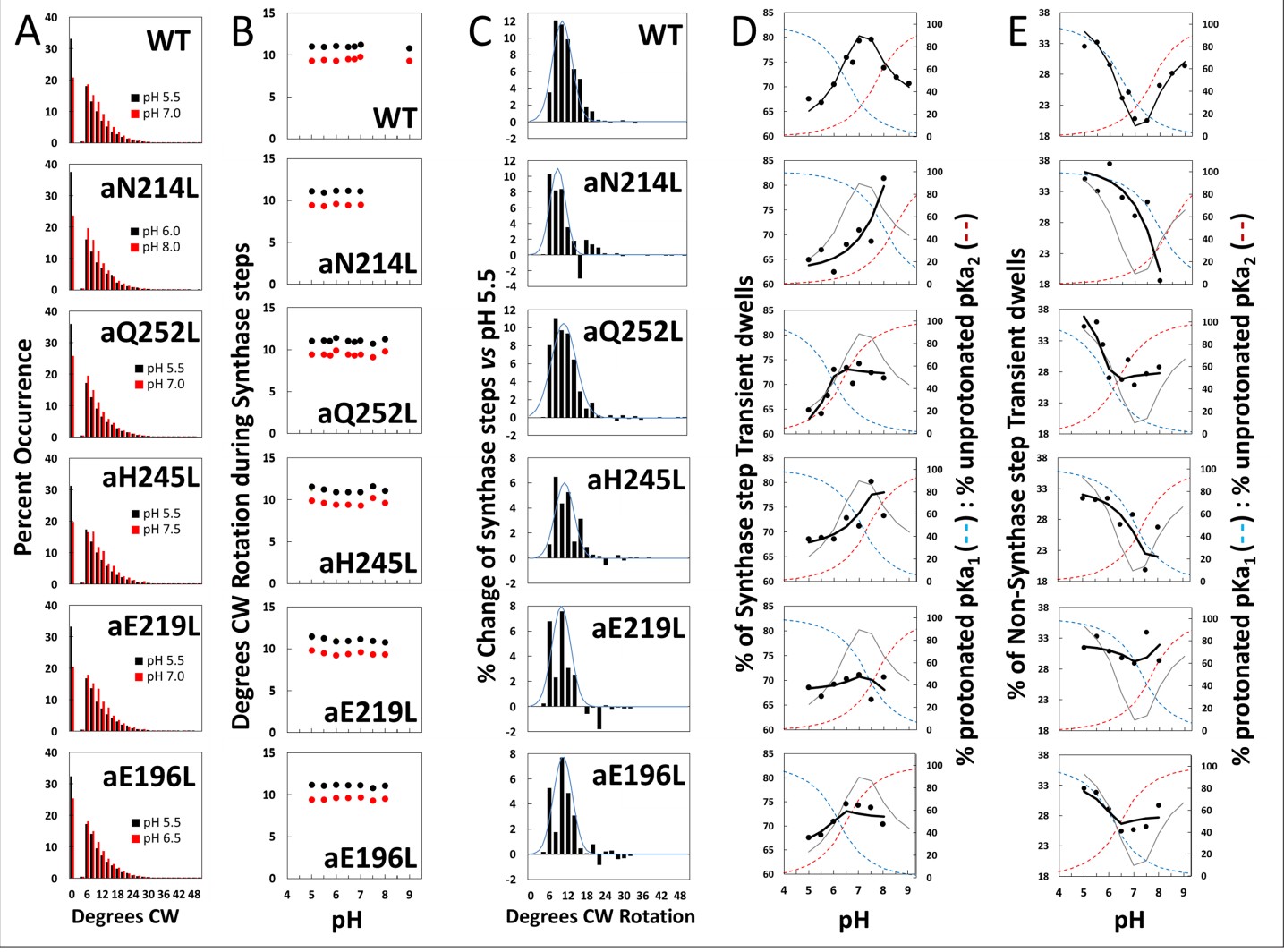

**Figure 4.** Effects of subunit-a mutations on the pH dependence of the extent of clockwise (CW) synthase-direction step rotation and fraction of transient dwells (TDs) containing synthase-direction steps. (**A**) Distributions of the extent of CW rotation in the ATP synthesis direction during transient dwells for WT and subunit-a mutants at the low (black) and high (red) pH values indicated. (**B**) Mean (black) and median (red) extents of CW rotation during a synthase-direction step versus pH. (**C**) Distributions of the difference in extent of CW synthase-direction step rotation between pH values in *Figure 2D* when the percent of synthase-direction steps was maximum versus minimum where the blue line is the Gaussian fit. (**D**) Percent of TDs containing CW synthase-direction steps versus pH, where the data were fit to *Equation 3* (black line). The fit for WT is shown as a gray line in the mutant plots. The fraction of protonated groups with $pKa_1$ (blue line), and unprotonated groups with $pKa_2$ (red line) versus pH was calculated from the pKa values of Table 2. (**E**) Percent of TDs that lack synthase-direction steps versus pH where the probability of forming a TD without a synthase-direction step (black line) was determined by *Equation 2* from the fraction of protonated groups with $pKa_1$ (blue line), and unprotonated groups with $pKa_2$ (red line) versus pH calculated using pKa values from Table 2. The fit for WT is shown as a gray line in the mutant plots.

The online version of this article includes the following figure supplement(s) for figure 4:

**Figure supplement 1.** Distributions of the extent of clockwise (CW) rotation in the ATP synthesis direction during transient dwells for WT and subunit-a mutants versus pH.

**Figure supplement 2.** Distributions of the difference in extent of clockwise (CW) synthase-direction step rotation between pH values when the percent of synthase-direction steps was maximum versus minimum, where (—) is the Gaussian fit.

of CW c-ring rotation during synthase events rotated 25° and 36° about 1% and 0.1% of the time, respectively.

## Subunit-a mutations affect the proportion of TDs with synthase-direction steps

We tested the hypothesis that synthase-direction steps result when both the input and output channels are in the correct protonation state to enable $H^+$ transfer to the carboxyl groups of the leading, and from the lagging c-subunits, respectively. If correct, the pH dependence of synthase-direction steps should follow the sum of the proportions of the protonated input and unprotonated output channels even when these pKa values change as the result of mutations. Alternatively, it was possible that the 11° rotations that we attribute to synthase-direction steps resulted instead from twisting of subunit-a and the c-ring as a single unit in response to the mismatch of the c-ring and catalytic dwell positions, which would not be subject to changes by subunit-a mutations.

The subset of TDs that forced the c-ring to rotate CW (synthase-direction steps) against the CCW force of $F_1$-ATPase rotation was pH dependent (**Figure 4D**). A maximum of 80% of TDs contained synthase-direction steps in WT at ~pH 7.3, and a minimum of 67% at pH 5.5. At pH values > 7.5, the proportion of synthase-direction steps decreased to 71% at pH 9.0.

Because a TD either contains ($T_S$) or lacks ($T_N$) a synthase-direction step, the pH dependence of TDs with a synthase-direction step (**Figure 4D**) was the inverse of that without a synthase-direction step (**Figure 4E**) per **Equation 2**.

$$T_S = 1 - T_N \tag{2}$$

For WT, the minimum $T_N$ of 20% at pH 7.5 increased 1.7-fold and 1.5-fold at pH 5.5 and at pH 9.0, respectively. At these extremes of low and high pH values, TD formation was dominated by groups where either $pKa_1$ is protonated or by unprotonated groups with $pK_2$. This conclusion is supported by the good fits of the pH dependencies of TDs without synthase-direction steps for WT and subunit-a mutants (**Figure 4E**) to **Equation 3**, where the probability of forming a TD without a synthase-direction step ($T_N$) is the sum of the probability ($P_1$) of the protonated group(s) with $pKa_1$ ($X_1$), and the probability ($P_2$) of unprotonated group(s) with $pKa_2$ ($Y_2$). Thus, these results support the conclusion that a TD without a synthase-direction step can result from a $H^+$ transfer event from the protonated group with $pKa_1$ *or* from a $H^+$ transfer event to the unprotonated group with $pKa_2$.

$$T_N = P_1(X_1) + P_2(Y_2) \tag{3}$$

Fits of the pH dependence of TDs without synthase-direction steps from **Equation 3** (black line) were based on the pKa values (**Figure 4E**), and probabilities summarized in **Table 2**. The WT data fit to probabilities of 38% and 33% for protonated groups ($pKa_1$ 6.5) and unprotonated groups ($pKa_2$ 7.7), respectively, such that the difference between the pKa values was 1.2 pH units. Consequently, $T_N$ showed a minimum at ~pH 7.3, and maxima at high and low pH values when only the group(s) with either $pKa_1$ or $pKa_2$ were protonated and unprotonated, respectively.

**Table 2.** pKa values and probabilities of forming transient dwells (TDs) without synthase-direction steps for WT and subunit-a mutants.
Values were derived from the fits of the data of **Figure 4C** to **Equation 2**.

|  | $pKa_1$ | $P_1$ (%) | $pKa_2$ | $P_2$ |
|---|---|---|---|---|
| WT | 6.5 | 38 | 7.7 | 33 |
| aN214L | 8.0 | 37 | 8.4 | 5 |
| aQ252L | 5.9 | 42 | 6.4 | 28 |
| aE219L | 7.1 | 32 | 7.4 | 35 |
| aH245L | 7.3 | 33 | 7.7 | 22 |
| aE196L | 6.2 | 34 | 6.5 | 28 |

As a result of the subunit-a mutations, $P_1$ values changed to a smaller extent (32–42%) than did $P_2$ values (5–35%). Except for aE219L, all mutations decreased $P_2$, including a >6-fold decrease with aN214L. The difference between pKa values observed with the mutants was from 0.3 to 0.5 pH units compared to the 1.2 pH unit difference of WT. Both $pKa_1$ and $pKa_2$ of aN214L increased by 1.5 and 0.7 pH units such that the minimum $T_N$ of ~18% at pH 8.0 represented an increase of 0.7 pH units from that of WT. At pH 5.5, $T_N$s comprised 38% of all TDs in aN214L. A similar but smaller shift of the minimum $T_N$ occurrence to pH 7.5 was also observed for aH245L, which primarily resulted from an increase in $pKa_1$ by 0.8 pH units from WT. A striking effect of mutations aQ252L,

aE219L, and aE196L was that they suppressed the pH dependence of synthase-direction step formation. Of these, aE219L was most suppressed where $T_S$ varied between 66% and 71% of TDs over the pH range examined.

In all cases, the occurrence of synthase-direction steps reached a maximum at the crossover point between the fractions of protonated groups with $pKa_1$ and unprotonated groups with $pKa_2$. This is the point at which the largest fractions of both groups were in the correct protonation state where $H^+$ transfer events could occur from the $pKa_1$ groups to the c-ring, *and* from the c-ring to the $pKa_2$ groups. These results eliminate the alternative hypothesis that synthase-direction steps result from the twisting of $F_O$ relative to $F_1$ as the result of elastic energy from $c_{10}$-ring:catalytic dwell mismatches because the elastic energy resulting from the mismatch in rotary positions would not be affected by these mutations.

## Discussion

The results presented here provide new insight into the mechanism by which the $F_O$ motor uses the energy from $H^+$ translocation to generate CW rotational torque on the c-ring to catalyze ATP synthesis. These studies support the hypothesis that residues associated with the half-channels work together to support a water column that transfers protons across the membrane in a coherent manner coupled to c-ring rotation in lieu of transferring protons directly and independently. These single-molecule investigations also tested the hypothesis that c-rotation operates via a Brownian ratchet versus a power stroke mechanism, and the results provide the first direct evidence that synthase-direction steps can occur by both mechanisms. Finally, the results presented here show that the proton translocation-dependent synthase-direction rotation occurs in 11° steps. These results do not support the hypothesis that the function of the essential aR210 is to deprotonate cD61 because recent $F_O$ structures show that the unprotonated lagging cD61 carboxyl is still 7.3 Å away from aR210 after an 11° c-ring rotation. Alternatively, an alternating two-step mechanism is proposed below to resolve this discrepancy.

### $F_O$ uses a Grotthuss mechanism to translocate protons through both half-channels

The results presented here support a Grotthuss mechanism in $F_O$ where water columns in each half-channel communicate via rotation-dependent $H^+$ transfer to and from the leading and lagging c-ring cD61 carboxyls. The coherent behavior of the water columns enables the release of a $H^+$ to the cytoplasm concurrent with each $H^+$ that enters the subunit-a input channel from the periplasm. This conclusion is supported by observations that: (i) ATP synthase-direction steps were maximal in WT and mutants when the fractions of protonated groups and unprotonated groups with low and high pKa values, respectively, were optimal for $H^+$ transfer both from the lagging cD61 to the output channel and from the leading cD61 to the input channel; (ii) mutation of a residue from either the input or output channel altered both the low and high pKa values of TDs indicating that the channels communicate; (iii) all mutations changed the ability to form TDs, indicating all the groups examined participate; and (iv) none of the mutations completely eliminated the ability to form TDs.

This conclusion is also consistent with the fact that participating residues aS199, aN214, and aQ252 are polar but not ionizable, and distances between channel residues that are too far apart for direct $H^+$ transfer but are positioned at distances able to support a water column. Although aQ252 is highly conserved (*Figure 5—figure supplement 1*), glycine or hydrophobic groups are naturally substituted: (i) for aN214 (*Toxoplasma gondii, Thermus thermophilus*); (ii) for aH245 (*Mycobacterium phlei, Tetrahymena thermophila, Acetobacter woodii, Toxoplasma gondii, Ilyobacter tartaricus, Fusobacterium nucleatum, Thermus thermophilus*); and (iii) for aE219 (*Tetrahymena thermophila, Euglena gracilis, Bacillus pseudofirmus OF4, Pichia angusta, Saccharomyces cerevisiae, Arthrospira platensis PCC9438*). Such mutations can be tolerated if the primary role of these groups was to support a water column that transferred protons.

A recent $F_1F_O$ structure from bovine mitochondria was of sufficient resolution to observe density near the input channel residues consistent with Grotthuss-type water molecules in this half-channel (*Spikes et al., 2020*). Unidentified electron densities near subunit-a input channel residues in $F_1F_O$ structures from *E. coli* (*Sobti et al., 2020*) and from *Polytomella* (*Murphy et al., 2019*) also suggest the presence of bound waters. The observation of a water column in both half-channels of the $V_O$

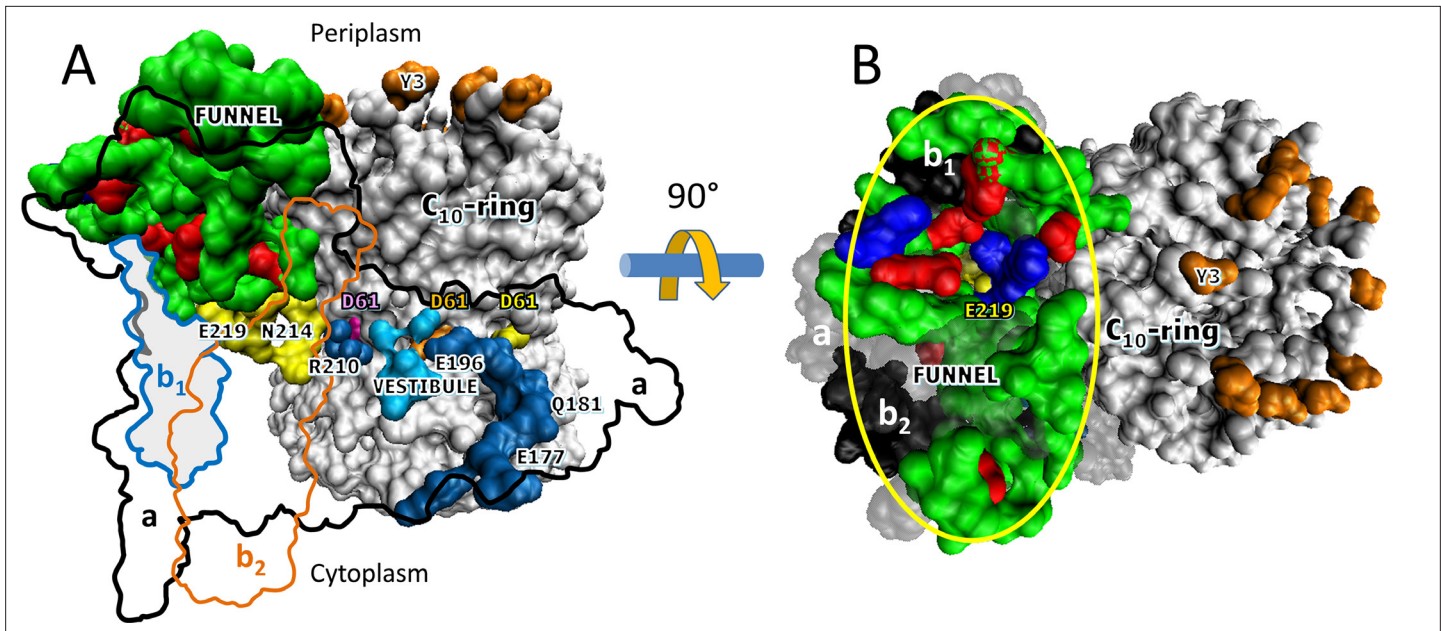

**Figure 5.** Aqueous funnel of charged and polar groups can serve as an antenna to supply protons to the input channel. (**A**) Transmembrane view of *Escherichia coli* $F_O$ (pdb-ID 6OQR) showing the path of charged and polar residues across the membrane. Outlines indicate space occupied by hydrophobic residues in subunit-a (black line), subunit-$b_1$ (blue line), and subunit-$b_2$ (orange line). The inner surface of the funnel, which is lined with polar residues and loop regions (green), acidic groups (red), and histidines (blue) from subunit-a and the subunit-$b_1$ N-terminus, is exposed to the periplasm at its wide end that narrows to aE219 (yellow) at the bottom. The input channel (yellow) extends from aE219 to aN214 and aQ252, which are proximal to aR210. Between aR210 and the output channel (dark blue) the leading (pink) and lagging (orange) cD61 groups rotate through a vestibule lined above and below the cD61 rotation plane by polar sidechains (light blue) that decrease the dielectric constant of the vestibule from that of the lipid bilayer. A protonated cD61 exposed to the lipid bilayer (yellow) is also visible. (**B**) Periplasmic surface of $F_O$ showing the interior surface of the funnel (orange oval) lined with charged and polar groups from subunit-a and subunit-$b_1$ as in **A** that narrows to aE219 (yellow) at the bottom where the input channel begins. Hydrophobic residues are shown of subunit-a (gray), subunits-$b_1$ and -$b_2$ (black), and the c-ring (white). The cY3 sidechains (orange) are shown to indicate the orientation of the periplasmic surface of the c-ring.

The online version of this article includes the following figure supplement(s) for figure 5:

**Figure supplement 1.** Sequence comparisons of the subunit-a helices that contact the c-ring.

complex (***Roh et al., 2020***) suggests that Grotthuss-based $H^+$ translocation is a commonly shared trait among the greater family of rotary ATPases.

Additional structural evidence that supports the existence of a Grotthuss $H^+$ translocation mechanism (***Figure 5***) is the presence of an ~30 Å diameter funnel that is lined with carboxylate and imidazole residues as the funnel narrows (***Figure 5***). The aE219-carboxyl examined here, which we propose to be the start of the Grotthuss column is positioned at the apex of this funnel. A Grotthuss mechanism was first proposed to explain extremely high rates of $F_O$-dependent $H^+$ translocation across *R. capsulatus* membranes (***Feniouk et al., 2004***). The rates were so fast that an ~40 Å diameter Coulomb cage of charged and polar groups was proposed to be required to serve as a $H^+$ 'antenna' to increase the delivery rate of protons from the aqueous solution to the entrance of the input channel water column (***Wraight, 2006***).

The results presented here indicate that multiple residues contribute to the pKa values that enable synthase-direction steps. The funnel extends the input channel from the residues investigated here that are located near the middle of the membrane to the periplasm, which provides the ultimate supply of protons to the c-ring. The charged and polar residues in the funnel are likely to contribute to the pKa value, and may also facilitate the protonation of the input channel relative to the output channel even though both half-channels are exposed to the same buffer as the result of the lipid bilayer nanodiscs employed. More work is necessary to address this issue.

## $F_O$ undergoes $H^+$ translocation-dependent 11° c-ring synthase-direction rotation steps

The extent of rotation during ATP synthase-direction steps was unexpectedly found to rotate CW by 11° in the WT and all mutants. Evidence presented here supports the conclusion that synthase-direction steps result from protonation of the leading cD61 from the input channel and from deprotonation of the lagging cD61 to the output channel to rotate the c-ring relative to subunit-a. These results include that: (i) synthase-direction steps depend on a group of residues with a low pKa that must be protonated, and a second group with a high pKa that must be unprotonated; (ii) formation of synthase-direction steps reached a maximum at the pH when the fractions of protonated groups with the low pKa and unprotonated groups with the high pKa were optimal; and (iii) mutating subunit-a residues in either the input or output half-channels altered both high and low pKa values, and altered the extent and pH dependence of synthase-direction step formation. The effects of the subunit-a mutants on the synthase-direction steps rule out the possibility that these steps result from twisting the entire $F_O$ relative to $F_1$.

During a CCW $F_1$-ATPase power stroke, TDs occur every 36°, which is equivalent to an interaction between subunit-a and each successive c-subunit in the *E. coli* $c_{10}$-ring. Since synthase-direction steps rotate by 11°, rotation by an additional 25° is required to advance the c-ring by one full c-subunit, which we observed in only 0.1% of the synthase-direction steps. Rotational sub-state structures (pdb-IDs 6OQR and 6OQS) of *E. coli* $F_1F_O$ that differ by a 25° rotation of the c-ring relative to subunit-a were obtained by cryo-EM (*Sobti et al., 2020*). Since advancing the c-ring by one c-subunit involves rotation by 36°, the difference between these sub-state structures also reveals information relevant to the 11° sub-step reported here.

The *E. coli* $F_1F_O$ rotational sub-state structures that differ by the 25° rotation of the c-ring relative to subunit-a were obtained when the complex was inhibited by ADP (*Sobti et al., 2020*). Similar 11° and 25° rotational sub-states have also been observed with ADP-inhibited $F_1F_O$ from *B. taurus* (*Zhou et al., 2015*) and from *M. smegmatis* (*Guo et al., 2021*). In *M. smegmatis* $F_1F_O$, the binding of bedaquiline stabilizes a rotational sub-state that is either 25° CW or 8° CCW from the equivalent rotational state in the absence of the drug (*Guo et al., 2021*). The rotational position of the c-ring in the cryo-EM structure of *S. cerevisiae* $F_1F_O$ is also changed by ~9° when the inhibitor oligomycin is bound to $F_O$ (*Srivastava et al., 2018*).

Several structural features of *E. coli* $F_O$ (*Sobti et al., 2020*) are relevant to its ability to undergo synthase-direction steps, and in combination with the results presented here, they provide insight into the mechanism of sustained CW rotation to power ATP synthesis. A transmembrane view of subunit-a (*Figure 5*) shows that aE196, aS199, aR201, aN214, and aQ252 are aligned along the plane of cD61 rotation. This plane is surrounded by hydrophobic residues that form a vestibule. Between aS199 and aR210, polar groups line the vestibule above (aS202 and aS206) and below (aK203 and aY263) the cD61 rotation plane. Although these polar groups do not directly participate in $H^+$ translocation (*Eya et al., 1991*), they enable water to access the vestibule (*Angevine and Fillingame, 2003*; *Angevine et al., 2003*) to make it less hydrophobic than the lipid phase of the membrane. Residues that provide a possible path for the output channel from aE196 to the cytoplasm include aQ181, aE177, and the subunit-a C-terminal carboxyl, which span this distance at ~4 Å intervals (*Sobti et al., 2020*), consistent with that needed to stabilize a Grotthuss water channel. More work is required to characterize this channel, especially since aE196 and aS199 are the only output channel residues conserved among other species (*Figure 5—figure supplement 1*).

A mechanism where $F_O$ uses alternating 11° (*Figure 6A,B*) and 25° (*Figure 6B,C*) sub-steps to power c-ring rotation that drives ATP synthesis is consistent with the data presented here, and with *E. coli* $F_1F_O$ structures 6OQR and 6OQS (*Sobti et al., 2020*). In 6OQS (*Figure 6A*), the lagging cD61 (orange) is 3.5 Å from aS199, which enables $H^+$ transfer to aS199 and aE196 via bound water. The leading cD61 (pink) is 3.8 Å from the aR210-guanidinium, consistent with intervening water. This cD61 is also proximal to aN214 and aQ252, which positions it for protonation from the input channel via bound water. In our model, the pH-dependent 11° sub-step (*Figure 6A*) occurs upon $H^+$ transfer from water bound to aN214 and aQ252 to the leading cD61, and $H^+$ transfer from the lagging cD61 to aS199 and aE196.

The low, medium, and high efficiencies of TD formation reported here (*Figure 2B*) were attributed to torsional strain resulting from the asymmetry between 36° $c_{10}$-ring stepping, and the 120 ° $F_1$ power

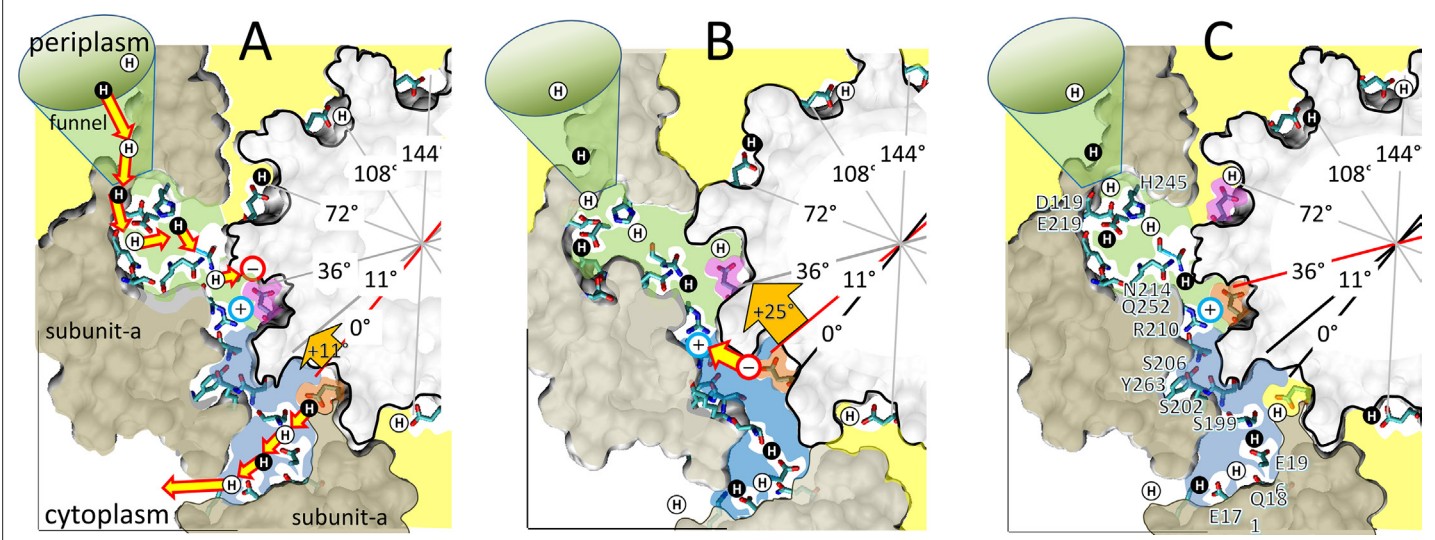

**Figure 6.** Alternating 11° and 25° sub-steps that power $F_O$ c-ring ATP synthase-direction rotation. (**A**) The pH-dependent 11° sub-step occurs when H⁺ transfer from aN214/aQ25-bound water to the unprotonated leading cD61-carboxyl (pink), and from the protonated lagging cD61-carboxyl (orange) to aS199/aE196-bound water. Upon displacement from aR210 by protonation, leading cD61 adopts the closed conformation to enable rotation into the lipid bilayer (yellow). Due to coherent H⁺ movement in the Grotthuss column, each H⁺ entering the input channel (green) from the funnel causes a H⁺ to exit the output channel (blue) to the cytoplasm. Rotation occurs when lagging cD61 is deprotonated because the negatively charged carboxyl moves in response to the decrease in hydrophobicity from the lipid bilayer to the water-containing vestibule (blue), and from the electrostatic attraction to aR210. This decreases the distance between the lagging cD61 carboxyl and the aR210-guanidinium from ~11.5 to ~7.5 Å. (**B**) The 25° sub-step occurs primarily from the electrostatic interaction between the lagging cD61 carboxy (orange) and the aR210-guanidinium. (**C**) Electrostatic attraction decreases the distance between orange cD61 and aR210 from ~7.5 to ~3.5 Å to complete a 36° stepwise c-ring rotation, which positions the orange cD61 to become the leading carboxyl for the next pH-dependent 11° sub-step. *Escherichia coli* $F_1F_O$ cryo-EM structures of rotary sub-states pdb-IDs 5OQS (**A and C**), and 5OQR (**B**) are shown as cross-sections of $F_O$ with hydrophobic resides of subunit-a (brown) and the c-ring (gray) along the plane defined by cD61 groups as viewed from the periplasm. Protons are alternately colored black and white to show the progression of proton transfer events.

strokes (*Yanagisawa and Frasch, 2017*; *Sielaff et al., 2019*). Based on this asymmetry observed in ADP-inhibited $F_1F_O$ structures (*Sobti et al., 2016*), high efficiency TD formation was proposed to occur (*Sielaff et al., 2019*) in the rotary state comparable to that in which rotary sub-state structures PDB-IDs 6OQR and 6OQS were subsequently observed at 3.1 Å resolution (*Sobti et al., 2020*). *Sobti et al., 2020*, concurred that torsional strain contributed to their ability to resolve the 6OQR and 6OQS sub-state structures. However, in results presented here, catalytically active $F_1F_o$ in lipid bilayer nano-discs show successive 11° ATP synthase-direction steps every 36° including at the rotary position of the ATPase power stroke where ADP inhibits rotation (*Figure 2A*, dashed line). Because ATP synthase-direction steps can also occur with low efficiency when torsional strain decreases the probability of forming a synthase-direction step, it is clear that torsional strain is not the primary contributing factor to the ability of $F_O$ to undergo 11° ATP synthase-direction steps.

After the 11° sub-step (*Figure 6B*), the distance of the now unprotonated lagging cD61 to the aR210-guanidinium decreased from ~11.5 to ~7.5 Å. These distances are inconsistent with the long-held belief that the role of aR210 is to displace the proton from the lagging cD61. Instead, we postulate that the electrostatic attraction between the negatively charged lagging cD61 and aR210 is sufficient to induce the 25° sub-step. As the result of this sub-step, the distance between these groups decreases from 7.5 to 3.8 Å (*Figure 6C*). As the 11° sub-step repeats, the subsequent loss of negative charge when the lagging cD61 is protonated by aN214 and aQ252 then allows this c-subunit to rotate away from aR210 into the lipid bilayer.

The probability that a TD occurs may appear to be stochastic. However, its occurrence depends on the kinetics and the energetics of the system. Slowing the angular velocity of the $F_1$ ATPase-driven power stroke increases TD occurrence at pH 8.0 (shown here to be suboptimal) indicating that the ability to form a TD depends on the rate that an interaction can form between subunit-a and each c-subunit relative to the angular velocity of $F_1$-ATPase-driven rotation (*Ishmukhametov et al., 2010*).

Evidence supports the hypothesis that the energy for $F_1$-ATPase power strokes is derived from ATP binding-dependent closure of the β-subunit lever domain upon subunit-γ, which is initiated at ~36° after the catalytic dwell in *E. coli* $F_1$ (*Martin et al., 2014*). Based on the $K_D$ of ATP at 36° measured in *Geobacillus stearothermophilus* $F_1$, the energy available for the power stroke from ATP binding is ~13.5 $k_BT$ (*Adachi et al., 2012*).

The results here suggest that the energy required to power the 11° synthase-direction step is close to that of the $F_1$-ATPase power stroke including: (i) that some synthase-direction steps oscillate consistent with a Brownian ratchet, especially those steps that occur late in the $F_1$ power stroke when the affinity for ATP is the highest; and (ii) that TD formation efficiency is increased or decreased (high and low efficiencies) by the 0.4 $k_BT$ of torsional energy from the ±14° rotary mismatch between $F_1$ and $F_O$ calculated from *Equation 4*, where θ is the rotational displacement in radians using the spring constant, κ, of 12.6 $k_BT$ radian$^{-2}$ measured for *E. coli* $F_1F_O$ (*Sielaff et al., 2008*).

$$U = 0.5\kappa\theta^2 \tag{4}$$

Consequently, the $F_O$ motor must have at least 13.5 $k_BT$ available to cause a TD. Possible sources of energy for TDs in addition to the 0.4 $k_BT$ of torsional energy include: (i) as much as 4.4 $k_BT$ from the difference of pKa values between the input and output channels; (ii) 5.9 $k_BT$ from the exclusion of the lagging charged cD61 carboxyl from the lipid bilayer into the aqueous vestibule, based on its measured desolvation energy (*White and Wimley, 1999*). The energy penalty of 0.8 $k_BT$ to insert the leading protonated cD61 carboxyl into the lipid bilayer is avoided by its conversion to the closed and locked position in the c-ring (*Pogoryelov et al., 2010*); and (iii) as much as 38.1 $k_BT$ from the electrostatic attraction of aR210 to unprotonated lagging cD61 at a distance of 11.5 Å when it is exposed directly to the lipid bilayer. The energy of this attractive force, which is highly dependent on the hydrophobicity of its environment, is calculated by the modified Coulomb equation (*Equation 5*), where $q_i$ and $q_j$ are elementary charges, $r_{ij}$ is the interatomic distance (in Ångstroms), and e is the dielectric constant, which is a measure of the hydrophobicity of the environment that ranges from 2 (lipid bilayer) to 80 (aqueous solvent). Although we do not yet know how wet the vestibule is, a dielectric constant of 13 and an 11.5 Å aR210-cD61 distance results in 3.8 $k_BT$, which when summed with the other energy sources totals 14.1 $k_BT$ without input of torsional energy. Since $F_1$-ATPase rotation from the catalytic dwell to the point that ATP binds is powered by no more than 4 $k_BT$ (*Martin et al., 2018*), this explains why synthase-direction rotation at these rotational positions typically has power stroke characteristics.

$$U = \frac{1}{e}\left(\frac{q_i q_j}{r_{ij}}\right) 561 \ \kappa_B T \tag{5}$$

After the 25° rotation step when the unprotonated cD61-aR210 distance is 3.8 Å, the electrostatic force is 11.4 $k_BT$ or 73.8 $k_BT$ when the dielectric constant is 13 or 2, respectively. Thus, the electrostatic interaction in a hydrophobic environment would be far too strong for any rotation to occur. More work is required to quantify the energetics of these sub-steps in the ATP synthesis mechanism because, when understood in combination with the steady-state pmf values and the dissociation constants of ATP, ADP, and Pi versus rotary position, these energy contributions will determine the non-equilibrium ATP/ADP•Pi concentration ratio that can be maintained by $F_1F_O$ at steady-state in vivo.

## Materials and methods
### Mutagenesis and purification of nanodisc $F_OF_1$
The $F_O$ subunit-a mutant plasmids were constructed from the pNY1-Ase plasmid containing the entire *unc* operon with a 6-His tag on the N-terminus of subunit-β and a cysteine inserted at the second position of subunit c (c2∇Cys), described previously by *Ishmukhametov et al., 2010*. The aN214L, aQ252L, aH245L, aE219L, and aE196L mutations were made on the plasmid using QuikChangesII-XL Site-Directed Mutagenesis Kit (Agilent). *E. coli* strain DK8, which lacks the *unc* operon, was transformed with the mutant plasmid. Cells were grown in 8 L of LB medium containing 50 µg/mL of ampicillin while shaking at 170 rpm at 37°C. About 40 g wet weight of cells was pelleted by centrifugation at 7700× *g* for 10 min at 4°C and stored at –80°C.

All subsequent steps were carried out at 4°C. The cell pellet was thawed, resuspended in 40 mL of French Press Buffer containing 200 mM Tris-HCl (pH 8.0), 100 mM KCl, 5 mM $MgCl_2$, 0.1 mM EDTA, and 2.5% (v/v) glycerol. The cells were lysed by running through a French Press at 12,000 psi, twice. Unbroken cells and cell debris were pelleted down at 7700× *g* for 15 min. The supernatant was transferred to an ultracentrifuge tube and the cell membrane was pelleted down by centrifugation at 180,000× *g* for 3 hr. To detergent-solubilize the $F_OF_1$, the membrane pellet was resuspended in Extraction buffer containing 6% (v/v) glycerol, 50 mM Tris-HCl (pH 8.0), 100 mM NaCl, 40 mM ε-aminocaproic acid, 15 mM *p*-aminobenzamidine, 1% octyl glucopyranoside, 0.5% sodium deoxycholate, 0.5% sodium cholate, 0.03% phosphatidylcholine, 30 mM imidazole, and 5 mM $MgCl_2$. The membrane suspension was incubated while gently shaking for 90 min and ultracentrifuged at 180,000× *g* for 2 hr. The supernatant was loaded into Ni-NTA column containing 1.5 mL Ni-NTA slurry that was equilibrated with Extraction buffer. The column was washed with 20 mL of Extraction buffer, and protein was eluted with 3 mL of Extraction buffer containing 200 mM imidazole. The protein concentration of the Ni-NTA column elution was determined by BCA assay and $F_OF_1$ was incorporated into lipid bilayer nanodiscs.

To form nanodiscs, membrane scaffold protein MSP-1E3D1 was used. The MSP was prepared by removing the His-tag with overnight TEV protease digestion at room temperature (at 25:1 ratio, w/w). Cleaved MSP was purified by passing through a Ni-NTA column. To incorporate the $F_OF_1$ into nanodiscs, a 1:4 molar ratio of $F_OF_1$:MSP was mixed with the addition of 10% stock sodium cholate in Extraction buffer to make the final sodium cholate concentration of 1%. To biotinylate the cysteine residue inserted at the N-terminus of subunit c, a 10-fold molar excess of biotin maleimide was added to $F_OF_1$. The mixture was gently shaken for 15 min to form biotinylated $F_OF_1$ nanodisc. The sample was desalted by running through a Sephadex G-50 column equilibrated with Buffer A containing 6% (v/v) glycerol, 50 mM Tris-HCl (pH 8.0), 100 mM NaCl, 4 mM *p*-aminobenzamidine, and 5 mM $MgCl_2$. The sample was aliquoted into 50 μL and quick frozen with liquid nitrogen until use. The presence of all $F_1F_O$ subunits and the MSP in each preparation was confirmed by SDS-PAGE.

### Gold-nanorod single-molecule experiments

Rotation of individual nanodisc $F_OF_1$ molecules were observed by single-molecule rotation assay. Sample slides were prepared with modifications of previously described methods (*Yanagisawa and Frasch, 2017*). Briefly, purified nanodisc $F_OF_1$ were immobilized on a microscope slide by the His-tag on subunit-β, unbound enzymes were washed off the slide with wash buffer (30 mM Tris, 30 mM PIPES, 10 mM KCl, at the appropriate pH), 80 × 40 nm AuNR coated with avidin was bound to the biotinylated c-ring of *E. coli* nanodisc $F_OF_1$, excess AuNRs were washed off with the wash buffer, and rotation buffer (1 mM $Mg^{2+}$ ATP, 30 mM Tris, 30 mM PIPES, 10 mM KCl, at the pH indicated) was added to the slide. The rotation of individual molecules was observed by measuring the change in intensity of polarized red light scattered from the AuNR using a single-photon detector. In each molecule observed, the rotation of the nanorod attached to an active nanodisc $F_OF_1$ complex was confirmed by the change in scattered light intensity as a function of the rotational position of the polarizing filter as described previously (*Spetzler et al., 2006*; *Hornung et al., 2011*). To make the measurement of nanodisc $F_OF_1$ undergoing power strokes, the orientation of the polarizing filter was adjusted to align with the minimum light intensity position that that corresponded to one of the three catalytic dwells. The sinusoidal change of polarized red light intensity was measured as the AuNR rotated from 0° to 90° relative to the catalytic dwell position. Measurements were taken in the form of 5 s data set at frame rate of 100 kHz. The occurrence of TDs in each subunit-a mutant was analyzed at varying pH from 5.0 to 8.0. TDs that occurred during the power strokes in the recorded data sets were analyzed by determining the $\arcsin^{1/2}$ of the intensity at each time point (*Martin et al., 2018*; *Martin et al., 2015 Sielaff et al., 2016*).

## Acknowledgements

This work was funded by NSF-BII 2119963, and from NIH R01GM097510 to WDF.

## Additional information

### Funding

| Funder | Grant reference number | Author |
|---|---|---|
| National Institute of General Medical Sciences | R01GM097510 | Wayne D Frasch |
| National Science Foundation | 2119963 | Wayne D Frasch |

The funders had no role in study design, data collection and interpretation, or the decision to submit the work for publication.

### Author contributions

Seiga Yanagisawa, Data curation, Formal analysis, Investigation, Visualization, Writing - original draft, Writing - review and editing; Wayne D Frasch, Conceptualization, Formal analysis, Funding acquisition, Methodology, Project administration, Supervision, Writing - original draft, Writing - review and editing

### Author ORCIDs

Wayne D Frasch (iD) http://orcid.org/0000-0001-6590-7437

### Decision letter and Author response

Decision letter https://doi.org/10.7554/eLife.70016.sa1
Author response https://doi.org/10.7554/eLife.70016.sa2

---

## Additional files

### Supplementary files

• Transparent reporting form

### Data availability

All data generated or analyzed during this study are included in the manuscript and supporting files. Data source files for all figures have been uploaded to DRYAD and can be located at: https://doi.org/10.5061/dryad.9cnp5hqhw.

The following dataset was generated:

| Author(s) | Year | Dataset title | Dataset URL | Database and Identifier |
|---|---|---|---|---|
| Yanagisawa S, Frasch WD | 2021 | Data from: pH-dependent 11° F1FO ATP synthase sub-steps reveal insight into the FO torque generating mechanism | http://dx.doi.org/10.5061/dryad.9cnp5hqhw | Dryad Digital Repository, 10.5061/dryad.9cnp5hqhw |

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
