## [Editor Report]

This manuscript describes single molecule spectroscopic work to probe the mechanisms by which protonation/deprotonation steps produce torque between the a-subunit and the C10 ring, which is subsequently conveyed to F1 to couple to ATP synthesis. This is an important bioenergetics question and the approach yields some tantalizing clues as to which protonation steps are involved. In principle, this knowledge could lead to direct experimental or computational tests to resolve the overall mechanism. The previous issues in the manuscript were well addressed by the authors in the new revision. The reviewers found a few small additional issues, which the authors can address in the final revisions.

---

## [Decision Letter]

**Decision letter after peer review:**

Thank you for submitting your article "pH-dependent 11{degree sign} F1FO ATP synthase sub-steps reveal insight into the FO torque generating mechanism" for consideration by *eLife*. Your article has been reviewed by 3 peer reviewers, and the evaluation has been overseen by David Kramer as Reviewing Editor and José Faraldo-Gómez as the Senior Editor. The following individual involved in review of your submission has agreed to reveal their identity: Werner Kühlbrandt (Reviewer #2).

The reviewers found the study to be an important, exciting and "elegantly conducted" contribution that begins to provide the "essential kinetic framework" needed to understand the recent, static high-resolution cryo-EM structures of F1FO ATPases. All reviewers, though, felt that the text itself was difficult to read, even for the expert, and that the issues would be extremely challenging for a more general audience. In some cases, these issues led to confusion relevant to the interpretation of the results and restructuring the text around a more hypothesis-driven framework should make these clearer. Finally, the Discussion seems to become defocused, including hypotheses that are tangential to the main points. a simplified, more focused revision would likely help the readers to understand the strongest points that are directly supported by the experimentation.

Essential revisions:

1) The Discussion should be restructured, focusing more on the component directly supported by the data presented. Relevant suggestions were made by all reviewers, but in particular the suggestion of Reviewer #1 that the text be given a more hypothesis-driven structure should help the reader to understand the context of the results. In other words, the text should make clear what the alternatives were and how the results favored the presented model.

2) A clarification of the relevance of dwell time to the overall kinetics needs to be included so that a more general audience can understand the critical functional connections. An additional scheme may help to achieve this.

3) The issue of flexibility of coupling between FO and F1 rotations, brought up by Reviewer #3, should be directly addressed, though additional experimentation is not needed.

4) The question raised by Reviewer #2 about the physicochemical impact of the formation of a salt bridge between aR210 of subunit-a and cD61 of the c-ring rotor should be directly addressed. How can such interactions promote (rather than hinder) rotation?

*Reviewer #1 (Recommendations for the authors):*

The text is quite dense and, in many places, difficult to read for the non-specialist, especially given the many references to specific structural details. See also the sentence beginning on line 152, and the paragraph starting on line 365. Scheme 5 is very useful in this respect and maybe it would be helpful to include a few more of such schemes. Specifically, a general scheme describing the expected impact of protonations on TD could help the reader contextualize the results.

Overall, the text could be far more impactful if it focused more tightly on the implications of the TD results themselves, testing specific sets of models, and taking more care to guide the readers through the interpretation. This could include describing the predictions of alternative hypothetical mechanisms, and how the TD results distinguish among them,

*Reviewer #2 (Recommendations for the authors):*

The introduction should mention explicitly that CCW rotation is powered by ATP hydrolysis. At present, this critical information is hidden in the discussion. The term "ATPase-driven" used in the introduction is not unambiguous, because rotary ATPases are both driven by ATP hydrolysis and drive ATP synthesis.

Figures and legends:

Figure 1: "HT" and the meaning of the light green bars are not explained in the legend.

Figure 2, line 148: The meaning of "each Slide2contained" is not evident.

Figure 5: The difference between the green and red protons in the figure is not explained.

Figure 6 A: aE219 (pink) is mostly obscured and barely visible. It is not visible at all in Figure 6B. Since aE219 seems to be in the middle of the aqueous vestibule rather than at its apex, it is not clear how it can channel protons to cD61. A schematic drawing might help. The dashed arrow in (B) that indicates the plane of cD61 rotation is yellow in the figure, but orange in the legend.

Essential text edits:

Line 90f: The sentence "These studies reveal that the CW rotation TDs occurs in pH-dependent 11{degree sign} ATP synthase sub-steps that depend… " is not intelligible as written. Probably a word is missing somewhere, and perhaps it should be "occur" rather than "occurs".

Line 410: The sentence "Transfer events between the c-ring one of the half channels result in TDs that lack a synthase step" does not make sense. Probably a word is missing.

Line 415 ff: The following sentence is not intelligible and needs sorting out: "Although this process is reversible, the results presented here that showed that ATP synthase steps increased with aQ252L and especially aN214L, decreased the efficiency of H^+^-transfer in the ATPase direction relative to the ATP synthase direction."

*Reviewer #3 (Recommendations for the authors):*

To strengthen this work, I would suggest the authors do the following:

i) Comment on the possibility that the rotations observed do not corollate with movements of the c-ring relative to subunit-a. Could the clockwise movements be some form of "spring back", with the 11 or 25 degree sub-steps showing the release of stored energy in the peripheral and/or central stalks. I cannot think of an easy way to test this hypothesis, other than completely redesigning the experiment to anchor subunit-a or changing the elastic properties of the stalks via mutagenesis. These would obviously be new studies, well beyond what is described in this manuscript.

ii) Gels confirming subunit-a is present, along with ATPase assays (with and without DCCD) in nanodiscs and proton pumping assays in liposomes should be provided for all the mutants in this study. The published literature shows similar mutants, however to my knowledge not all positions are mutated to Leucine as they are in this study.

iii) Comment on the activity of the mutants relative to wild type. Are there any differences in ATPase activity between the mutants? Could the frequency or length of transient dwells explain any changes seen in ATPase activity?

iv) Comment on the what causes the transient dwells. Is there a way to show that these are related to F1-ATPase inhibition? Could the authors add Azide or AMP-PNP to increase the likelihood of inhibition?

v) Overlay successive transient dwell power strokes from the same molecule at the same rotatory position. It would be interesting to see whether the same molecule consistently performs 11 or 25 degree movements at the same rotary position. This could be potentially used to disseminate whether the CW sub steps are caused by "spring back" from the peripheral stalk or synthesis sub steps.

vi) Modify the text: The introduction is quite confusing to the non-specialist reader. The discussion is longer than the results and the methods are very brief.

---

## [Author Response]

Essential revisions:1) The Discussion should be restructured, focusing more on the component directly supported by the data presented. Relevant suggestions were made by all reviewers, but in particular the suggestion of Reviewer #1 that the text be given a more hypothesis-driven structure should help the reader to understand the context of the results. In other words, the text should make clear what the alternatives were and how the results favored the presented model.2) A clarification of the relevance of dwell time to the overall kinetics needs to be included so that a more general audience can understand the critical functional connections. An additional scheme may help to achieve this.3) The issue of flexibility of coupling between FO and F1 rotations, brought up by Reviewer #3, should be directly addressed, though additional experimentation is not needed.4) The question raised by Reviewer #2 about the physicochemical impact of the formation of a salt bridge between aR210 of subunit-a and cD61 of the c-ring rotor should be directly addressed. How can such interactions promote (rather than hinder) rotation?

All completed as requested.

Reviewer #1 (Recommendations for the authors):The text is quite dense and, in many places, difficult to read for the non-specialist, especially given the many references to specific structural details. See also the sentence beginning on line 152, and the paragraph starting on line 365. Scheme 5 is very useful in this respect and maybe it would be helpful to include a few more of such schemes. Specifically, a general scheme describing the expected impact of protonations on TD could help the reader contextualize the results.

We have now added additional explanation sentences for the non-specialist. We also revised the structure figures extensively to increase clarity.

Overall, the text could be far more impactful if it focused more tightly on the implications of the TD results themselves, testing specific sets of models, and taking more care to guide the readers through the interpretation. This could include describing the predictions of alternative hypothetical mechanisms, and how the TD results distinguish among them,

We revised the manuscript substantially with this goal in mind.

Reviewer #2 (Recommendations for the authors):The introduction should mention explicitly that CCW rotation is powered by ATP hydrolysis. At present, this critical information is hidden in the discussion. The term "ATPase-driven" used in the introduction is not unambiguous, because rotary ATPases are both driven by ATP hydrolysis and drive ATP synthesis.

Clarified as requested.

Figures and legends:Figure 1: "HT" and the meaning of the light green bars are not explained in the legend.

Changed as requested.

Figure 2, line 148: The meaning of "each Slide2contained" is not evident.

Corrected.

Figure 5: The difference between the green and red protons in the figure is not explained.

Red and green protons were used to enable the reader to see that the protons have moved from frame to frame. A sentence was added at the end of the Figure legend to explain this about the protons that are now indicated by black and white.

Figure 6 A: aE219 (pink) is mostly obscured and barely visible. It is not visible at all in Figure 6B. Since aE219 seems to be in the middle of the aqueous vestibule rather than at its apex, it is not clear how it can channel protons to cD61. A schematic drawing might help. The dashed arrow in (B) that indicates the plane of cD61 rotation is yellow in the figure, but orange in the legend.

We have now revised Figures 6 and 7 to clarify this and other points.

Essential text edits:Line 90f: The sentence "These studies reveal that the CW rotation TDs occurs in pH-dependent 11{degree sign} ATP synthase sub-steps that depend… " is not intelligible as written. Probably a word is missing somewhere, and perhaps it should be "occur" rather than "occurs".

Corrected.

Line 410: The sentence "Transfer events between the c-ring one of the half channels result in TDs that lack a synthase step" does not make sense. Probably a word is missing.

Corrected.

Line 415 ff: The following sentence is not intelligible and needs sorting out: "Although this process is reversible, the results presented here that showed that ATP synthase steps increased with aQ252L and especially aN214L, decreased the efficiency of H^+^-transfer in the ATPase direction relative to the ATP synthase direction."

Corrected.

Reviewer #3 (Recommendations for the authors):To strengthen this work, I would suggest the authors do the following:i) Comment on the possibility that the rotations observed do not corollate with movements of the c-ring relative to subunit-a. Could the clockwise movements be some form of "spring back", with the 11 or 25 degree sub-steps showing the release of stored energy in the peripheral and/or central stalks. I cannot think of an easy way to test this hypothesis, other than completely redesigning the experiment to anchor subunit-a or changing the elastic properties of the stalks via mutagenesis. These would obviously be new studies, well beyond what is described in this manuscript.

We have now added text that refers to the Introduction and the Results that identify how the results of the experiments rule out the possibility of the alternate hypotheses proposed by the reviewer.

ii) Gels confirming subunit-a is present, along with ATPase assays (with and without DCCD) in nanodiscs and proton pumping assays in liposomes should be provided for all the mutants in this study. The published literature shows similar mutants, however to my knowledge not all positions are mutated to Leucine as they are in this study.

We added information to the Introduction that cites our previous work where we showed the gels confirming that subunit-a is present (Ishmukhametov et al. EMBO J 2010), as well as the DCCD results that the reviewer has requested. As requested, we added a paragraph to the Introduction that refers to previous work describing the effects of the mutants to leucine studied here regarding their effects on ATP hydrolysis activity and proton pumping rate.

iii) Comment on the activity of the mutants relative to wild type. Are there any differences in ATPase activity between the mutants? Could the frequency or length of transient dwells explain any changes seen in ATPase activity?

The results of the single-molecule studies presented here do not conflict with the effects of mutations on activity reported previously.

iv) Comment on the what causes the transient dwells. Is there a way to show that these are related to F1-ATPase inhibition? Could the authors add Azide or AMP-PNP to increase the likelihood of inhibition?

We added information to the Introduction and to the Discussion to address the reviewer’s concerns. Addition of azide or AMP-PNP would not, in our opinion, provide any more information than the DCCD experiment that we already published.

v) Overlay successive transient dwell power strokes from the same molecule at the same rotatory position. It would be interesting to see whether the same molecule consistently performs 11 or 25 degree movements at the same rotary position. This could be potentially used to disseminate whether the CW sub steps are caused by "spring back" from the peripheral stalk or synthesis sub steps.

For the present manuscript, we feel that we have provided evidence that clearly demonstrates that the CW sub steps are not an artifact caused by “spring back” from the peripheral stalk but instead represent rotation of the c-ring relative to subunit-a, and that no further experiments are required to prove this point further. In addition, we present evidence that the “spring back” from the peripheral stalk does impact the efficiency of TD formation as we reported previously and has been corroborated by other labs.

The reviewer does pose a very interesting question that additional analyses to see whether the same molecule consistently performs 11 or 25 degree movements at the same rotary position could provide additional insight into the F1Fo mechanism, but such studies are beyond the scope of the current manuscript, which already represents 4 years of work.

vi) Modify the text: The introduction is quite confusing to the non-specialist reader. The discussion is longer than the results and the methods are very brief.

We added a substantial amount of background information to the Introduction to address the reviewer’s concern. We also added information to the Results to help the non-specialist readers understand the purpose of the experiments and the reasons that the evidence supports the conclusions, and we added additional details to the Methods.